# Satellite-based meteorological drought indicator to support food security in Java Island

Siswanto Siswanto[1], Kartika Kusuma Wardani[2,3], Babag Purbantoro[4], Andry Rustanto[2], Faris Zulkarnain[2,3], Evi Anggraheni[5], Ratih Dewanti[3], Triarko Nurlambang[2,3,6,7,8], Muhammad Dimyati[2]*

1 Center for Applied Climate Information and Services, Agency for Meteorology, Climatology, and Geophysics (BMKG), Jakarta, Indonesia, 2 Department of Geography, Faculty of Mathematics and Natural Science, Universitas Indonesia, Depok, Indonesia, 3 Center for Applied Geography Research, Universitas Indonesia, Depok, Indonesia, 4 Remote Sensing Technology and Data Centre, Indonesian Institute of Aeronautics and Space (LAPAN), Jakarta, Indonesia, 5 Department of Civil and Environmental Engineering, Faculty of Engineering, Universitas Indonesia, Depok, Indonesia, 6 Research Center for Climate Change Universitas Indonesia, Depok, Indonesia, 7 Disaster Risk Reduction Center Universitas Indonesia, Depok, Indonesia, 8 Sustainable Development Goals Hub Universitas Indonesia, Depok, Indonesia

* m.dimyati@sci.ui.ac.id

**Data Availability Statement:** The data that support the findings of this study were obtained from (1) ESRI ArcGIS base map for imagery data (Fig 1), (2) OpenStreetMap for shapefile data (Fig 1), (3) the

## Abstract

A meteorological drought refers to reduced rainfall conditions and is a great challenge to food security. Information of a meteorological drought in advance is important for taking actions in anticipation of its effects, but this can be difficult for areas with limited or sparse ground observation data available. In this study, a meteorological drought indicator was approached by applying the Standardized Precipitation Index (SPI) to satellite-based precipitation products from multiple sources. The SPI based meteorological drought analysis was then applied to Java Island, in particular to the largest rice-producing districts of Indonesia. A comparison with ground observation data showed that the satellite products accurately described meteorological drought events in Java both spatially and temporally. Meteorological droughts of the eight largest rice-producing districts in Java were modulated by the natural variations in El Niño and a positive-phase Indian Ocean Dipole (IOD). The drought severity was found to be dependent on the intensity of El Niño and a positive-phase IOD that occurs simultaneously, while the duration seems to be modulated more by the positive-phase IOD. The results demonstrate the potential applicability of satellite-based precipitation monitoring to predicting meteorological drought conditions several months in advance and preparing for their effects.

## Introduction

In Indonesia, various regions have reported drought events indicated by a scarcity of surface water [1, 2], which may be followed by depletion of the shallow groundwater table [3] and reduced crop productivity [4, 5]. These events are mainly induced by the high seasonality of

Indonesian Agency for Meteorology, Climatology, and Geophysics (BMKG) for SA-OBS data (Figs 2 to 4), (4) NASA's Earth Data for TRMM data (Figs 2 to 4), (5) Center for Hydrometeorology and Remote Sensing (CHRS) for PERSIANN data (Figs 2 to 4), (6) Climate Hazards Center UCSB for CHIRPS data (Figs 2 to 4), and (7) Ministry of Agriculture Republic Indonesia for rice production data. All the satellite imagery data that were used in this study (Figs 2 to 4) were from open-source resources. The authors confirm that the data supporting the findings of this study are available within the following URLs. The URLs provide precipitation data (raw data) where each dataset was downloaded. The SPI data (processed data), authors upload together in this Supporting information ZIP file. SA-OBS: https://sacad. database.bmkg.go.id/ TRMM: https://disc.gsfc. nasa.gov/mirador-guide PERSIANN: https:// chrsdata.eng.uci.edu/ CHIRPS: https://data.chc. ucsb.edu/products/CHIRPS-2.0/.

**Funding:** This research was financially supported by National Research Priority (PRN) Ministry of Research and Technology/National Research and Innovation Agency (RISTEK/BRIN) and Indonesia Endowment Fund for Education (LPDP under grant number: 250/E1/PRN/2020. The funders had no role in study design, data collection and analysis, decision to publish, or preparation of the manuscript.

**Competing interests:** The authors have declared that no competing interests exist.

the rainfall, which is particularly erratic for regions located farthest south from the Equator such as Java, Nusa Tenggara, and the southern parts of Sumatra, Kalimantan, and Papua [6, 7]. Drought events in Indonesia are closely related to global climate phenomena, particularly El Niño [6, 8] and the Indian Ocean Dipole (IOD) [8]. The strong El Nino 1997/1998 impacted on the declining 1998 paddy harvest in Indonesia by 3.6 percent below 1997 and 6 percent below the 1996 harvest, due to the worst drought in that decade, according to the FAO report [9]. Also during 2015 and early 2016, a strong El Niño has suppressed rainfall over a wide part of the Indonesian archipelago. Rainfall in Indonesia's prime rice growing areas on the islands of Java, Sumatra, Sulawesi, and Kalimantan has been erratic and deficient, forcing farmers to delay planting well beyond the optimal period during the wet season which traditionally runs from October through April [10]. A strong El Niño may also induce intense fires in forest and peatland areas [11] that cause transboundary haze or smoke. There is a need for sustained monitoring and accurate forecasting of meteorological conditions to anticipate and minimize the negative effects of drought events.

Droughts can be classified as meteorological, agricultural, hydrological, or socioeconomic. At present, Indonesia has no services that provide comprehensive drought information to the public. The Indonesian Agency for Meteorology, Climatology, and Geophysics (BMKG) only provides a meteorological drought warning service. A meteorological drought refers to a temporary reduction in the amount of precipitation compared to the average [12, 13]. The degree of the drought is determined according to the ratio between the amount of precipitation and the amount considered normal for that period, and the duration is determined by how long the dry condition persists. A meteorological drought affects the agricultural sector by causing a groundwater deficit, which can result in crop failure, and reducing the water reserves of dams available for irrigation. If a meteorological drought persists, a hydrological drought may occur, which is defined as a reduced supply of surface water and groundwater and is based on the measured water levels of rivers, reservoirs, lakes, and groundwater [14]. BMKG defines a meteorological drought as an area experiencing dry conditions for a certain time due to reduced precipitation or a longer dry season than normal, and it issues warnings based on the number of consecutive days without rain (i.e., dry spell) and prediction of a low precipitation amount. Predicting dry spells is very important because they have a major influence on agriculture, especially rice production [15, 16]. Prolonged dry spells can reduce the yield [17] and even the overall rice production [18]. Dry spells are the most sensitive indicator of the effect of El Niño on Indonesia [19].

BMKG monitors for dry spell based meteorological droughts on a 10-day timescale, which is complemented by daily dry spell monitoring derived from near-real-time the daily precipitation product estimated from Global Satellite Mapping of Precipitation (GSMaP). However, the utilization of available multisource or selected satellite precipitation products is an ongoing challenge that needs to be addressed to improve the comprehensiveness, speed, and accuracy of drought information services in the spatial and temporal domains. Using satellite products for near- and post-real-time precipitation estimation can greatly improve the quality of drought analysis. Several high-resolution precipitation products derived from meteorological satellites with worldwide coverage have recently become freely available in near real time with a horizontal resolution of 0.05˚–0.25˚ [20–23]. These include Tropical Rainfall Measuring Mission (TRMM) and its successor Global Precipitation Measurement (GPM), GSMaP, Climate Hazards Group InfraRed Precipitation with Stations (CHIRPS), and Precipitation Estimation from Remotely Sensed Information using Artificial Neural Networks–Climate Data Record (PERSIANN-CDR).

Several studies have evaluated satellite products for the best fit to ground observations of precipitation and meteorological drought. CHIRPS showed the closest agreement to rain

gauge data compared with PERSIANN-CDR and TRMM for the Bundelkhand region of central India [24]. PERSIANN-CDR and CHIRPS were found to be more accurate than TRMM for meteorological drought monitoring of Chile in South America [25]. PERSIANN-CDR performed better than CHIRPS for semiarid regions of Brazil [26]. GPM, GSMaP, and PERSIANN were all found to offer reliable near-real-time meteorological estimation of data-sparse regions in Iran [27]. Climate Prediction Center Morphing Method (CMORPH) outperformed TRMM and Integrated Multisatellite Retrievals for Global Precipitation Measurement (IMERG) for daily estimation of the precipitation in the Yellow River basin of China, while GSMaP and CHIRPS performed the worst. These satellite products have demonstrated an acceptable accuracy compared to rain gauge data in some areas of Indonesia, although there have been some difficulties and differences, particularly in the representation of extremely heavy rain conditions [28–31]. CHIRPS was found to correlate well with rain gauge data from six meteorological stations installed by BMKG with a correlation coefficient of 0.70–0.86 [32]. CHIRPS outperformed GSMaP and IMERG at assessing the monthly precipitation of Bali Island [31]. TRMM performed better than Asian Precipitation–Highly-Resolved Observational Data Integration Towards Evaluation (APHRODITE) for the Pemali–Comal River basin [33].

In this study, we analyse a meteorological drought indicator that applies the widely used Standardized Precipitation Index (SPI) to precipitation products from multiple satellite data sources. This differs from currently available where the SPI is calculated based on sparse surface observation data. The evaluation of multi-source SPI with El Niño and IOD simultaneously is expected to provide a better, optimal, and more accurate representation and understanding of meteorological drought events in the study area. We used the SPI based meteorological drought indicator to evaluate the satellite precipitation products against ground observation data for Java Island, with a focus on eight of the 10 largest rice-producing districts as the national food barn of Indonesia. These eight largest rice-producing districts are located in Java Island, namely Karawang, Subang, Indramayu that are located in West Java Province, Cilacap, Grobogan, Sragen are located in Central Java. Ngawi and Lamongan are located in East Java Province (Fig 1). The paddy fields and rice yields of the eight districts in 2019 are shown in Table 1. In these regions, rice is cultivated mostly in lowland areas, with crops typically being irrigated, well watered and heavily fertilized. Rainfed rice fields that only rely on rainwater are also found in this area. Water-related disasters, such as flood and drought, will negatively affect rice production areas and yields, which can threaten food security. A study by Pratiwi et al. (2020) revealed that in Central Java, the 2014 flood event affected 94,306 hectares (ha) paddy fields, while drought in 2015 affected 82,324 ha paddy fields [10].

The eight districts still have a relatively low density of rain gauges and other surface meteorological observation networks. Accurate information and forecasting of meteorological drought conditions for these districts is fundamental for food security by allowing countermeasures to be performed early in advance. This study may also help address the effects of climate change on Java, which has experienced a continuous decrease in precipitation during the dry season over the last three decades (1981–2010) [35].

## Data and methods

### Data

We used monthly precipitation estimated from four datasets covering Java with similar spatial resolutions. Three datasets were satellite-based precipitation products: CHIRPS, TRMM, and PERSIANN. The Southeast Asia Observational Dataset (SA-OBS) was used to validate the

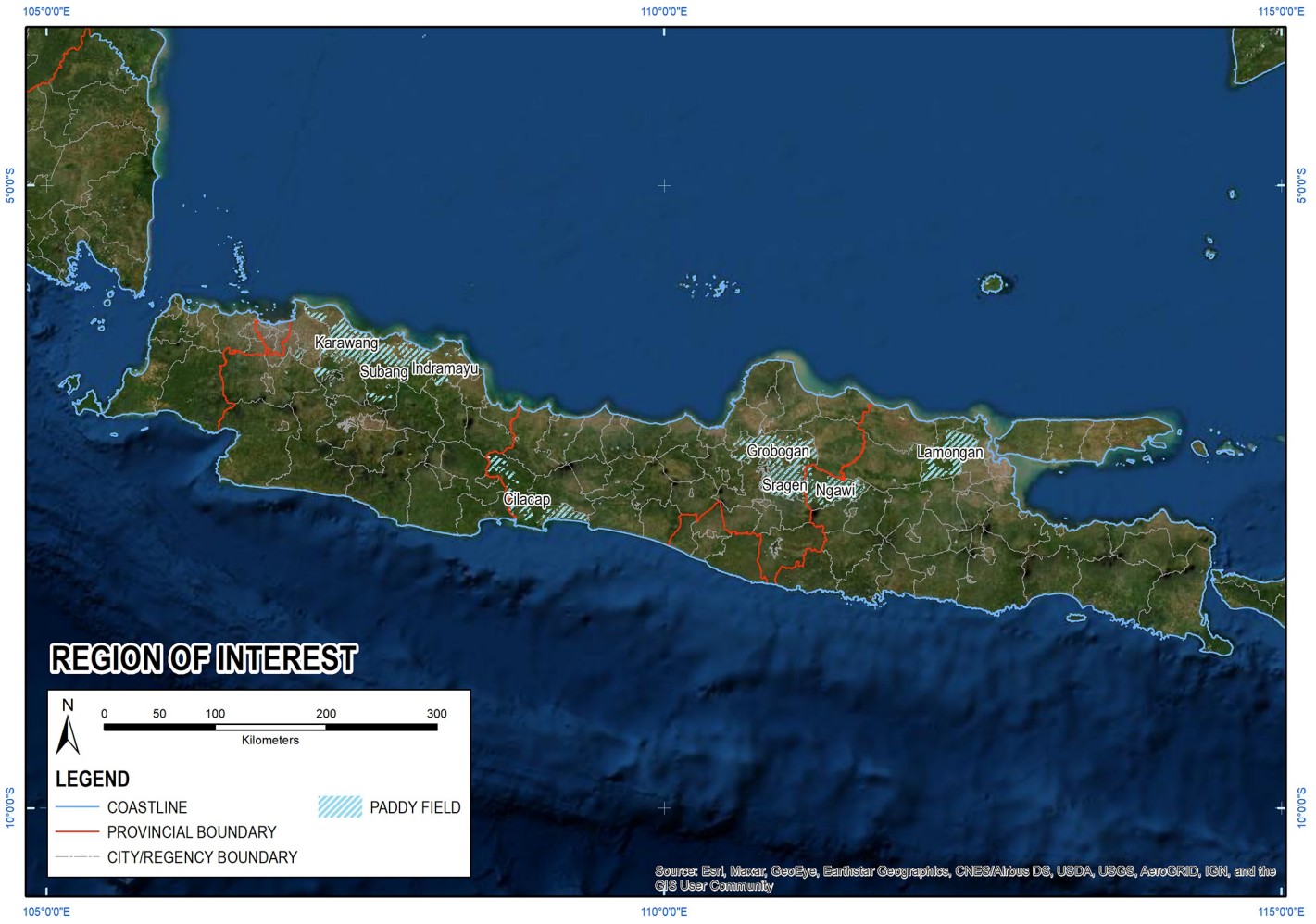

**Fig 1. Study area: Java Island and the eight districts of the largest rice producer in Indonesia as the region of interest.** The greener areas on the map indicate paddy fields of the region of interest.

satellite products. Table 2 summarizes the datasets, their resolutions, and time covered. The datasets are briefly described below.

**Climate Hazards Group Infrared Precipitation with Stations.** CHIRPS is a satellite product with multiple data sources [36] that is available at high resolution (0.05˚ × 0.05˚) and

**Table 1. The rice fields and yields of the eight districts in 2019 [34].**

| District | Rice (Paddy) Fields (ha) | Rice Production (tones-brutto) | Unhulled Rice Production (tones-netto) |
|---|---|---|---|
| Indramayu | 215.731 | 1.376.429,68 | 789.657,71 |
| Karawang | 185.807 | 1.117.814 | 641.290 |
| Subang | 156.298,50 | 942.932 | 540.960 |
| Lamongan | 140.463,58 | 839.724 | 481.750 |
| Ngawi | 122.500,97 | 777.190 | 445.874 |
| Grobogan | 136.209,59 | 772.521 | 443.196 |
| Sragen | 111.569,05 | 766.012 | 439.461 |
| Cilacap | 439.461,26 | 699.965 | 401.570 |

**Table 2. Summary of the datasets used in this study.**

| Data source | Temporal resolution | Spatial resolution | Time period |
|---|---|---|---|
| CHIRPS | Monthly | 0.05˚ × 0.05˚ | January 1981–January 2021 |
| TRMM | Monthly | 0.25˚ × 0.25˚ | January 1998–January 2020 |
| PERSIANN | Monthly | 0.25˚ × 0.25˚ | March 2000–April 2021 |
| SA-OBS | Monthly | 0.25˚ × 0.25˚ | January 1981–December 2017 |

a long time series from 1981 to the present [37]. Regional evaluations have shown that estimations using CHIRPS agree well with ground observations from local gauge networks, which is promising for drought monitoring [38]. In this study, we used the CHIRPS v.2 Global Monthly dataset for the period March 1981–January 2021. The CHIRPS dataset used in this study is obtained from https://data.chc.ucsb.edu/products/CHIRPS-2.0/.

**Tropical Rainfall Measuring Mission.** TRMM has three types of data products: orbital (also known as swath), gridded, and TRMM-related (e.g., ancillary products, ground-based instrument products, TRMM and ground observation subsets, and field experiment products) [39]. In this study, we used a gridded data product: the TRMM (TMPA) 3B43 V7 Monthly dataset for the period March 1998–January 2020. TMPA 3B43 is a widely used dataset because of its high spatial and temporal resolutions. The latest version (Version 7) includes a uniform data reprocessing and calibration scheme and a single use of Global Precipitation Climatology Centre monthly rain gauge analysis, which improves the accuracy compared to previous versions [30]. The TMPA 3B43 dataset used in this study is downloaded from https://disc.gsfc.nasa.gov/mirador-guide.

**Precipitation Estimation from Remotely Sensed Information using Artificial Neural Networks.** PERSIANN is based on geostationary infrared imagery and daytime visible imagery. This system was first developed in 2000, and it estimates surface precipitation rates based on local cloud texture characteristics by using infrared images with an artificial neural network (ANN) [40, 41]. The ANN generates global precipitation for the latitudes of 60˚ S to 60˚N in two steps: an automatic clustering process is applied to transforming infrared (wavelength of 10.2–11.2 μm) images into a hidden layer to form a self-organizing feature map (SOFM); then the discrete SOFM clusters in the hidden layer are made continuous [42, 43]. The PERSIANN dataset used in this study is downloaded from https://chrsdata.eng.uci.edu/.

**Southeast Asia Observational Dataset.** SA-OBS is a newly available high-resolution land-only gridded dataset for Southeast Asia. It provides the daily precipitation amount and daily minimum, mean, and maximum temperatures. It was developed by BMKG and the Royal Netherlands Meteorological Institute (KNMI) with the aim of providing an observational basis against which model results can be compared [44]. SA-OBS can be compared against existing gridded datasets based either on a less dense network of stations covering the same area or on satellite data where the interpretation of infrared and microwave signals as precipitation or temperature has limited accuracy, such as APHRODITE [45, 46], CMORPH [47], and TRMM [48]. We used a SA-OBS dataset with a regular 0.25˚ × 0.25˚ grid resolution for the period of January 1981–December 2017. The dataset is available freely for research purposes and can be downloaded from https://sacad.database.bmkg.go.id/.

We also used monthly data from the Oceanic Nino Index (ONI) and Dipole Mode Index (DMI) to monitor the activities of the El Niño–Southern Oscillation (ENSO) in the Pacific Ocean and the IOD in the Indian Ocean. The ONI was obtained from the National Weather Service Climate Prediction Center (https://origin.cpc.ncep.noaa.gov/), while the DMI was retrieved from the Global Climate Observing System (https://psl.noaa.gov/).

## Methods

We used the SPI to determine meteorological drought conditions in the study area. The SPI is commonly used for meteorological drought analysis and can be calculated for various time scales to evaluate not only the water supply in the short term but also the available water resources in the long term [49, 50]. McKee et al. developed the SPI in 1993 based on their understanding that a precipitation deficit has different effects on the groundwater, reservoir storage, soil moisture, snowpack, and streamflow [12, 51]. The SPI is computed by measuring precipitation anomalies at a given location, which are identified by comparing the observed total precipitation with the long-term historical record for a period of interest (e.g., 1, 3, 12, or 48 months) [52]. The historical record is fitted to a gamma probability distribution, which is then transformed into a normal distribution such that the mean SPI for the given location and period is zero. The SPI is calculated as follows [49]:

$$x_s^m(t) = \sum_{i=g-s+1}^{g} x(i), g = 12(t-1) + m \qquad (1)$$

where $x$ is the monthly precipitation series, $t$ is the yearly index, and $m$ is a specific month (e.g., January, February).

For a given location, a rainfall deficit (i.e., meteorological drought) occurs when the SPI is less than –1.0, while excess rainfall occurs when the SPI is greater than 1.0 (Table 3). Because the SPI is given in units of standard deviation from the long-term mean, it can be used to compare precipitation anomalies for any geographic location and any number of timescales. In this study, we used the 3-month (SPI-3) and 6-month (SPI-6) timescales. SPI-3 compares the precipitation over a specific 3-month period with the precipitation over the same 3-month period for all years included in the dataset; it reflects short- and medium-term moisture conditions and provides a seasonal estimation of precipitation, which is useful for evaluating the available water supply for primary agricultural regions [53]. Similarly, SPI-6 compares the precipitation over a specific 6-month period with the precipitation over the same 6-month period for all years included in the dataset. SPI-6 indicates seasonal to medium-term trends in precipitation and may also be associated with anomalous streamflows and reservoir levels depending on the location and time of year [53, 54].

In this study, we calculated the SPI by using the open-source module Climate and Drought Indices in Python (https://climate-indices.readthedocs.io/en/latest/) [55]. All SPI analyzes and plots were performed using the open source platform NCAR Command Language (https://www.ncl.ucar.edu) [56] and R (https://www.R-project.org) [57], while the selected paddy field areas and study domain were drawn using ArcGIS (https://www.esri.com) [58].

**Table 3. Drought severity based on SPI values [53].**

| | |
|---|---|
| 2.0+ | Extremely wet |
| 1.5 to 1.99 | Very wet |
| 1.0 to 1.50 | Moderately wet |
| −0.99 to 0.99 | Near normal |
| −1.0 to −1.50 | Moderately dry |
| −1.5 to −1.99 | Severely dry |
| −2 and less | Extremely dry |

## Results

### Accuracy of the satellite products at describing precipitation climatology

Fig 2 shows the annual total precipitation (left), mean monthly precipitation during the peak of the rainy season (middle), and mean monthly precipitation during the dry season (right) of

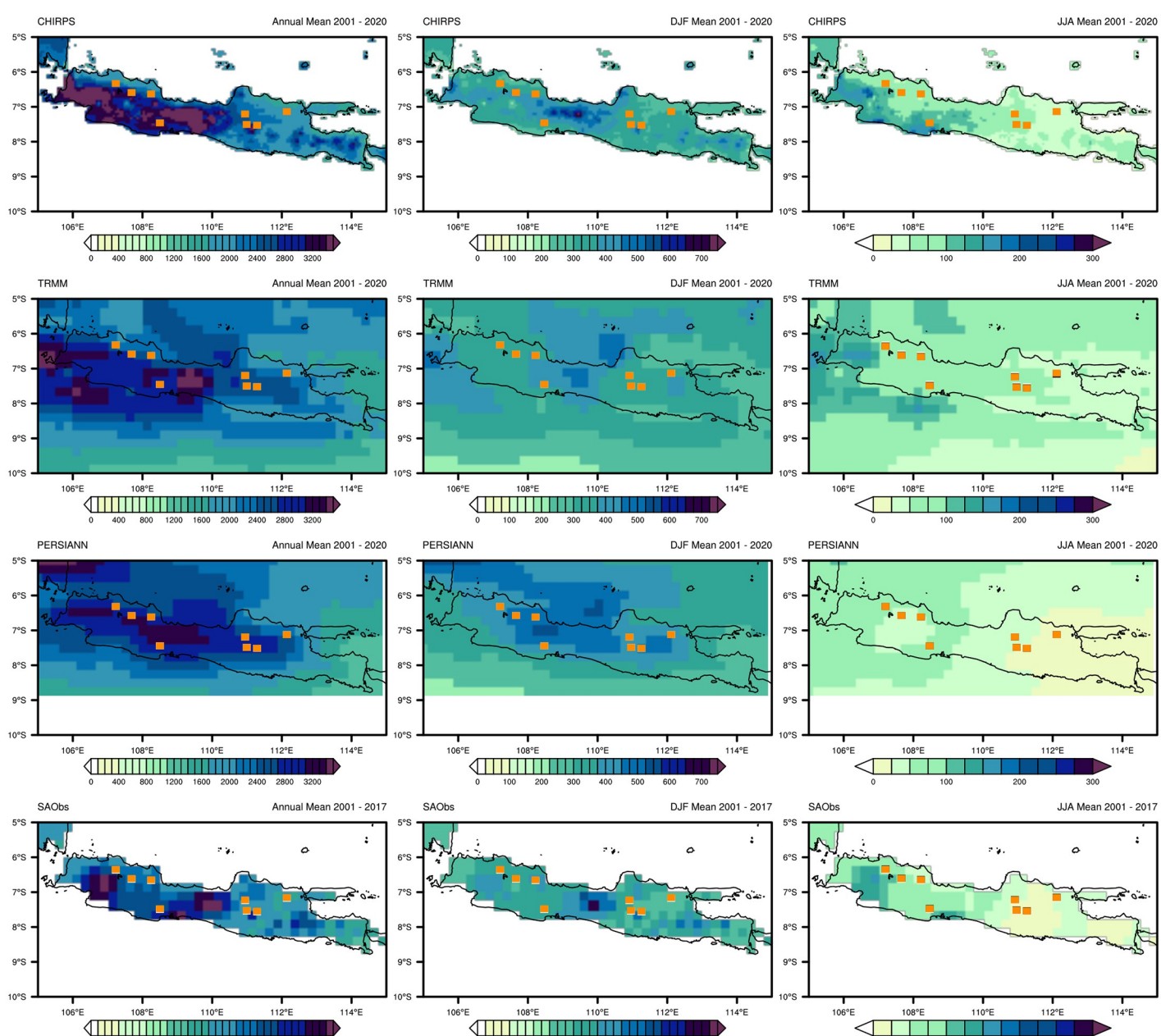

**Fig 2. Comparison of the annual total precipitation (left), mean monthly precipitation in the rainy season (middle), and mean monthly precipitation in the dry season (right) over the study area according to 20 years of climatological data from CHIRPS, TRMM, PERSIANN, and SA-OBS.** The rainy season corresponds to December, January, and February (DJF); the dry season corresponds to June, July, and August (JJA). Dots on the map indicate the largest rice-producing districts of Indonesia.

Java according to the satellite products (CHIRPS, TRMM, PERSIANN) and observational dataset (SA-OBS) for 2001–2020. We analyzed the original datasets without ocean masking or changing the horizontal resolution. The three satellite products returned similar and comparable results for the total annual precipitation that agreed with SA-OBS; in general, West Java was wetter than other parts of the study area. CHIRPS and TRMM accurately described the parts of the study area with the highest precipitation intensity (>3200 mm/year): the southern part of West Java and central part of Central Java. However, PERSIANN did not return to such a high intensity. These parts of the study area with a high annual precipitation are dominated by highlands or mountains. PERSIANN also showed a wider distribution of precipitation over the study area but failed to describe the parts with the highest precipitation. TRMM also identified a different location with the highest precipitation: the southern part of West Java. In general, the precipitation tended towards areas close to the sea. Seven of the rice-producing districts had an annual precipitation of 2000–2510 mm/year, while Cilacap received even more. The variation in annual precipitation is generally determined by the precipitation variations at the peaks of the rainy and dry seasons. The peak of the rainy season is defined as the months with the highest precipitation intensity and frequency. In contrast, the peak of the dry season is defined as the months with the smallest reduction in the precipitation amount and frequency. For Java, the peaks of the rainy and dry seasons are typically December–February (DJF) and June–August (JJA), respectively.

During the peak of the rainy season, almost all of the datasets indicated that the central mountainous part of Central Java had the highest precipitation intensity in the study area. PERSIANN slightly differed by indicating a wider distribution area. CHIRPS, TRMM, and SA-OBS showed good agreement on the precipitation distribution for the study area, with particularly strong agreement between CHIRPS and SA-OBS. During this period, the eight rice-producing districts generally received about 251–400 mm/month of precipitation.

At the peak of the dry season, the precipitation intensity for the entire study area was generally 0–151 mm/month. All datasets indicated that West Java was wetter and East Java was drier with varying spatial distributions. PERSIANN and SA-OBS agreed on the dry conditions of East Java, while CHIRPS and TRMM indicated slightly wetter conditions. CHIRPS, TRMM, and SA-OBS agreed on the wetter conditions of West Java. All rice-producing districts in West Java and Cilacap received monthly precipitation of about 100 mm/month, while rice-producing districts in eastern Central Java and East Java experienced very low monthly precipitation of <51 mm/month. The peak of the rainy season corresponds to December, January, and February (DJF); the peak of the dry season corresponds to June, July, and August (JJA).

## Accuracy of the satellite products at describing drought events: Spatial and temporal domains

In Java, drought conditions are generally influenced by El Niño and a positive-phase IOD. Drought events strongly affect agricultural productivity in Java. We needed to determine the reliability of satellite-based precipitation products at describing the spatial and temporal distributions of drought events. As an example, we applied the SPI to describe the spatial distribution of drought conditions in Java for a drought event that occurred in mid-2015 that was caused by a strong El Niño in the Pacific Ocean and positive-phase IOD in the Indian Ocean. Fig 3 shows the SPI-3 results for the spatial distribution of the meteorological drought event in June (left), August (middle), and October (right). All datasets showed that the drought started from the western part of Java in June. CHIRPS and SA-OBS agreed that the drought was uniformly distributed in West Java, but TRMM and PERSIANN indicated that the drought was more concentrated on the north shore. SA-OBS had high absolute values for SPI-3 of less than

## Standardized Precipitation Index (SPI)

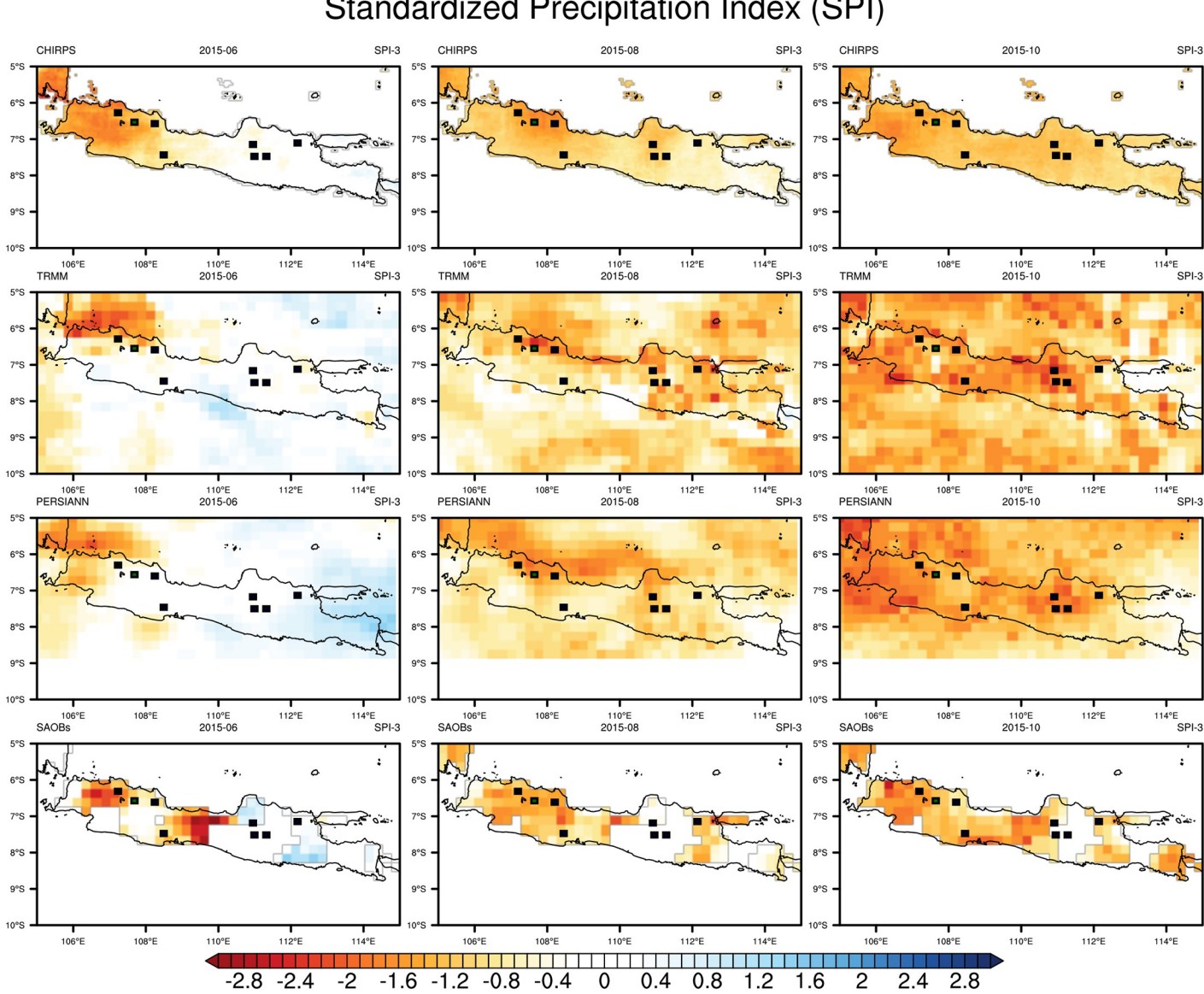

**Fig 3. SPI-3 derived from CHIRPS, TRMM, PERSIANN, and SA-OBS for June (left), August (middle), and October 2015 (right) during the drought event due to the 2015–16 El Niño.** Dots on the map indicate the largest rice-producing districts of Indonesia.

−2.4 in some parts of the study area while the surrounding parts had SPI values of more than −1.2. SA-OBS had blank SPI-3 values for some parts of the study area, which was attributed to a possible lack of observations for that time. No significant drought was detected in Central Java and East Java in June. The largest rice-producing districts in West Java and the western part of Central Java were affected by meteorological drought conditions based on the SPI being between −1.2 and −2.0 (moderately to very dry). All datasets showed a gradual spatial expansion of intense meteorological drought conditions toward the east of the study area in August and October. The SPI values of TRMM, PERSIANN, and SA-OBS indicated that very dry conditions had spread over almost all of Java by August, including the eight largest rice-producing districts, and extremely dry conditions (below −2) had spread in some parts of West Java by October. Because of its higher spatial resolution, CHIRPS gave a smoother

## Standardized Precipitation Index (SPI)

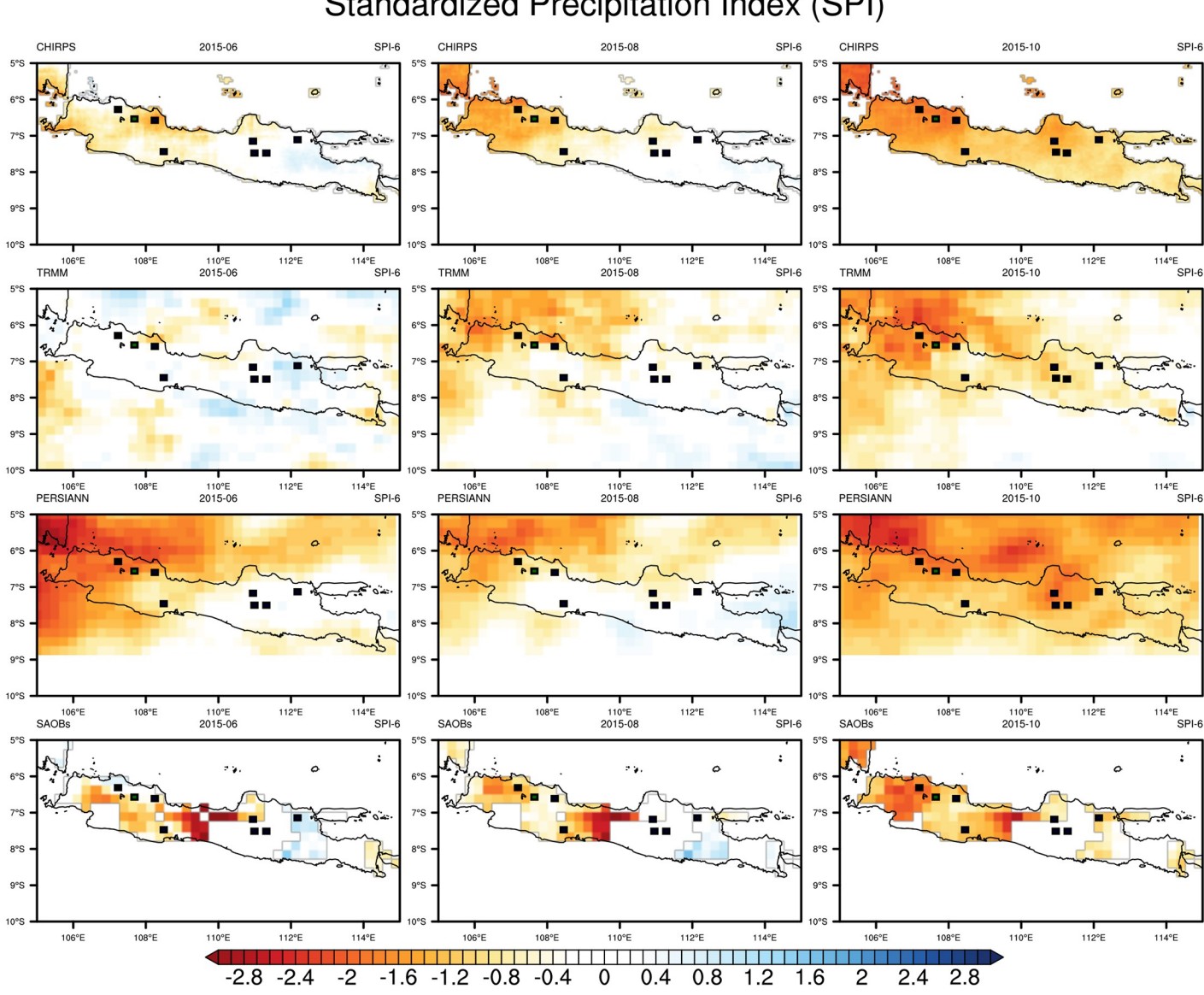

**Fig 4. Six-month SPI (SPI-6) derived from CHIRPS, TRMM, PERSIANN, and SA-OBS for June 2015 (left), August 2015 (middle), and October 2015 (right) during the drought event due to the 2015–16 El Niño.** Dots on the map indicate the largest rice-producing districts of Indonesia.

distribution than the other datasets. SA-OBS returned a rougher distribution in some areas because of the sparse observational data.

Similar to Figs 3 and 4 presents the spatial distribution of SPI-6 values over Java for June (left), August (middle), and October 2015 (right). Compared to CHIRPS and TRMM, SA-OBS and PERSIANN indicated drier conditions in June, while all datasets indicated a comparable expansion of drought conditions in August. From June, SA-OBS indicated moderate to very dry conditions (SPI-6 of −1 to −2) over the western half of Java and extremely dry conditions (SPI-6 of less than −2) in Central Java. All datasets indicated that the meteorological drought had worsened over all of Java by October. Moreover, most of the eight largest rice-producing districts were suffering from the meteorological drought with very dry to extremely dry conditions. The occurrence of an El Niño appears to trigger anomalies in the river flow and reservoir

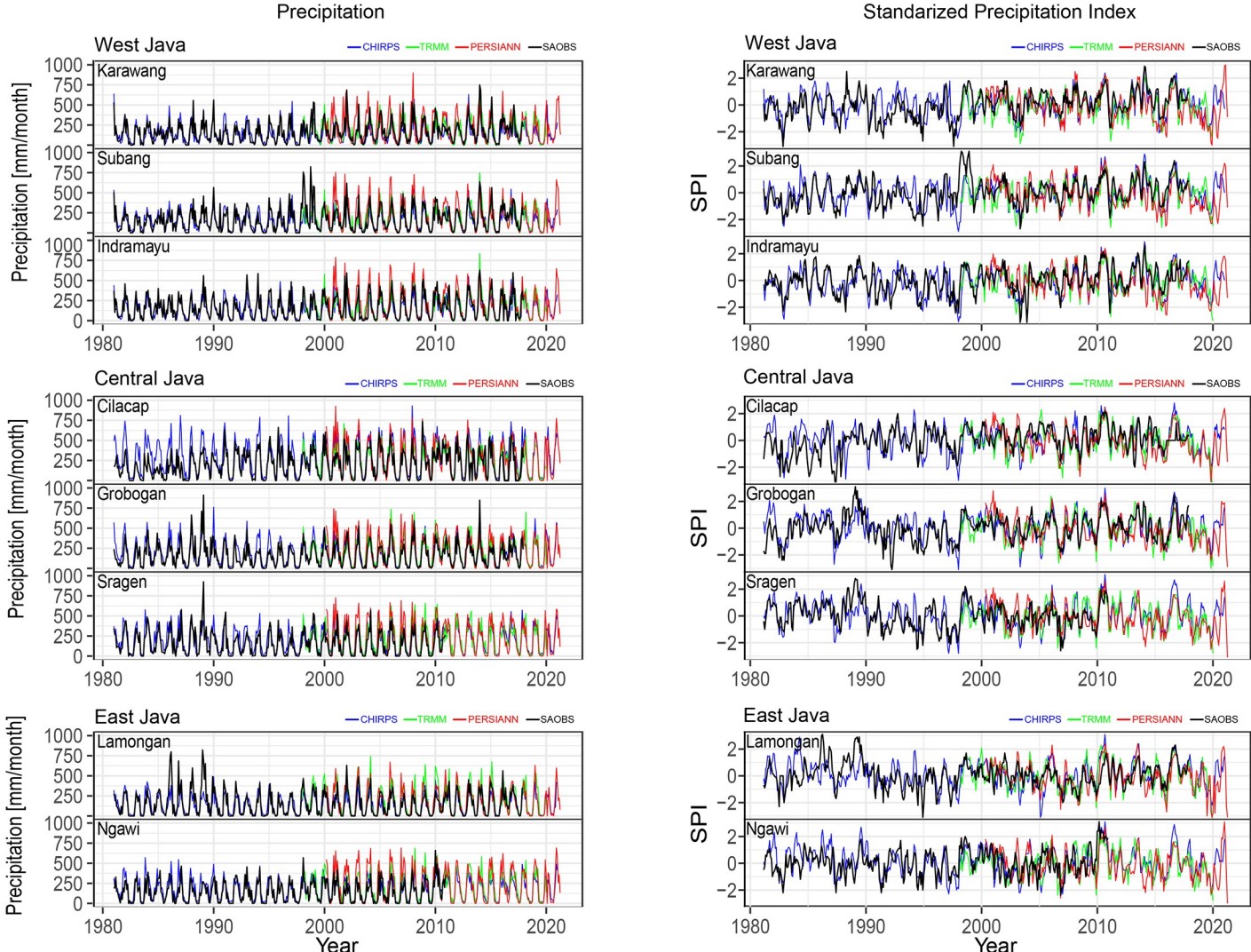

**Fig 5. Time series for the monthly precipitation (left) and SPI-3 (right) of the eight largest rice-producing districts in Java grouped by province: West Java (top), Central Java (middle), and East Java (bottom).** The monthly precipitation and SPI-3 were determined from CHIRPS (blue), TRMM (green), PERSIANN (red), and SA-OBS (black).

levels of the study area 5–6 months later; this can be used as an early warning for the agricultural sector to take action against potential drought events.

The reliability of the satellite products also needed to be evaluated in the temporal domain to determine the suitability of the SPI as a tool for drought monitoring. We focused on the drought characteristics for the eight largest rice-producing districts of Java: Karawang, Subang, and Indramayu in West Java Province; Cilacap, Grobogran, and Sragen in Central Java Province; and Ngawi and Lamongan in East Java Province. Fig 5 depicts the time series of the monthly precipitation (left) and SPI-3 (right) of these districts grouped by province: West Java (top), Central Java (middle), and East Java (bottom). The monthly precipitation and SPI-3 were derived from CHIRPS (blue line), TRMM (green line), PERSIANN (red line), and SA-OBS (black line). The time series were obtained for all available periods covered by the datasets. In general, the four datasets showed similar evolutions and variations in precipitation and drought conditions for all locations. Differences were observed in the monthly

precipitation intensity; for example, TRMM or PERSIANN sometimes indicated a higher or lower monthly precipitation intensity than the other datasets. Consequently, SPI-3 values from TRMM and PERSIANN sometimes indicated drier or wetter conditions than the other datasets. For certain locations (e.g., West Java and Central Java), PERSIANN sometimes exhibited large deviations or even the opposite trend from the other datasets, which may be due to its algorithms.

For a quantitative view, Fig 6 evaluates the accuracy of the monthly precipitation and SPI-3 derived from satellite products in terms of their proximity to the observational data, which is indicated by the correlation R and root mean square (RMSE). We assumed that the SA-OBS dataset is the ground truth and validated the satellite products against them. The satellite products were concluded to be valid if the data distribution was centered or converged to the 1:1 reference line, and the slope of the regression line closely fit the reference line. We only used the period of 2001–2017 so that all datasets could be compared on the same basis. CHIRPS and TRMM showed higher correlations with SA-OBS than PERSIANN with R values of 0.82 and RMSE around 0.7 for the monthly precipitation and >0.6 for SPI-3. For a longer time window (1998–2017), CHIRPS had a higher correlation a lower RMSE to SA-OBS than TRMM (not shown).

## Meteorological drought characteristics in the study area

As noted previously, drought in Indonesia has traditionally been related to global climate phenomena, particularly the ENSO in the Pacific Ocean. However, research on the relation of drought to the IOD is still limited [8]. Fig 7 presents a parallel time series of the ONI and DMI showing the significance of the ENSO and IOD, respectively, as inter-annual climate drivers of the precipitation variability in Indonesia. We wanted to determine if a temporal relationship exists between variations in monthly precipitation and SPI-3 and these phenomena that can be used as an indicator of meteorological drought. We built a multisource averaged SPI from individual SPIs derived from CHIRPS, TRMM, PERSIANN, and SA-OBS. Most of the individual SPIs showed good agreement, except for those from PERSIANN for the last part of the time series. The multisource averaged SPI showed that severe meteorological droughts always occurred in conjunction with a strong El Niño or positive-phase IOD. Strong El Niño events (ONI > +2) occurred in 1982–83, 1997–98, and 2015–16. Other El Niño events were weak or moderate. These strong El Niño events were always followed by meteorological drought events that could be extreme. The peaks of these extreme droughts occur simultaneously with the peaks of ONI with no time lag. The seasonal precipitation patterns persisted even though the intensity was lower than usual during these events. Meteorological droughts with very dry conditions also occurred in 1987, 1992, 1994–95, 2007, and 2015 following weak to moderate El Niño events. The meteorological droughts in response to weak and moderate El Niño events did not occur simultaneously, and the droughts were moderately to very dry. In 1992, the very dry condition showed a stronger correlation with the IOD timeline rather than El Niño, which peaked later.

For the SPI results, the colored bar shows the average from multiple sources, and the lines show the values derived from individual datasets: CHIRPS (blue), TRMM (green), PERSIANN (red), and SA-OBS (black). For each time series, the vertical orange solid line indicates a year with El Niño, and a vertical light-blue line indicates a year with La Niña. The horizontal gray dashed lines indicate the levels of ENSO (top), precipitation (middle), and SPI (bottom).

Interestingly, the 2015 drought event only reached very dry conditions even though a strong El Niño took place at the time. The positive-phase IOD was not very strong for this event, so the meteorological drought did not reach extremely dry conditions as in the cases of

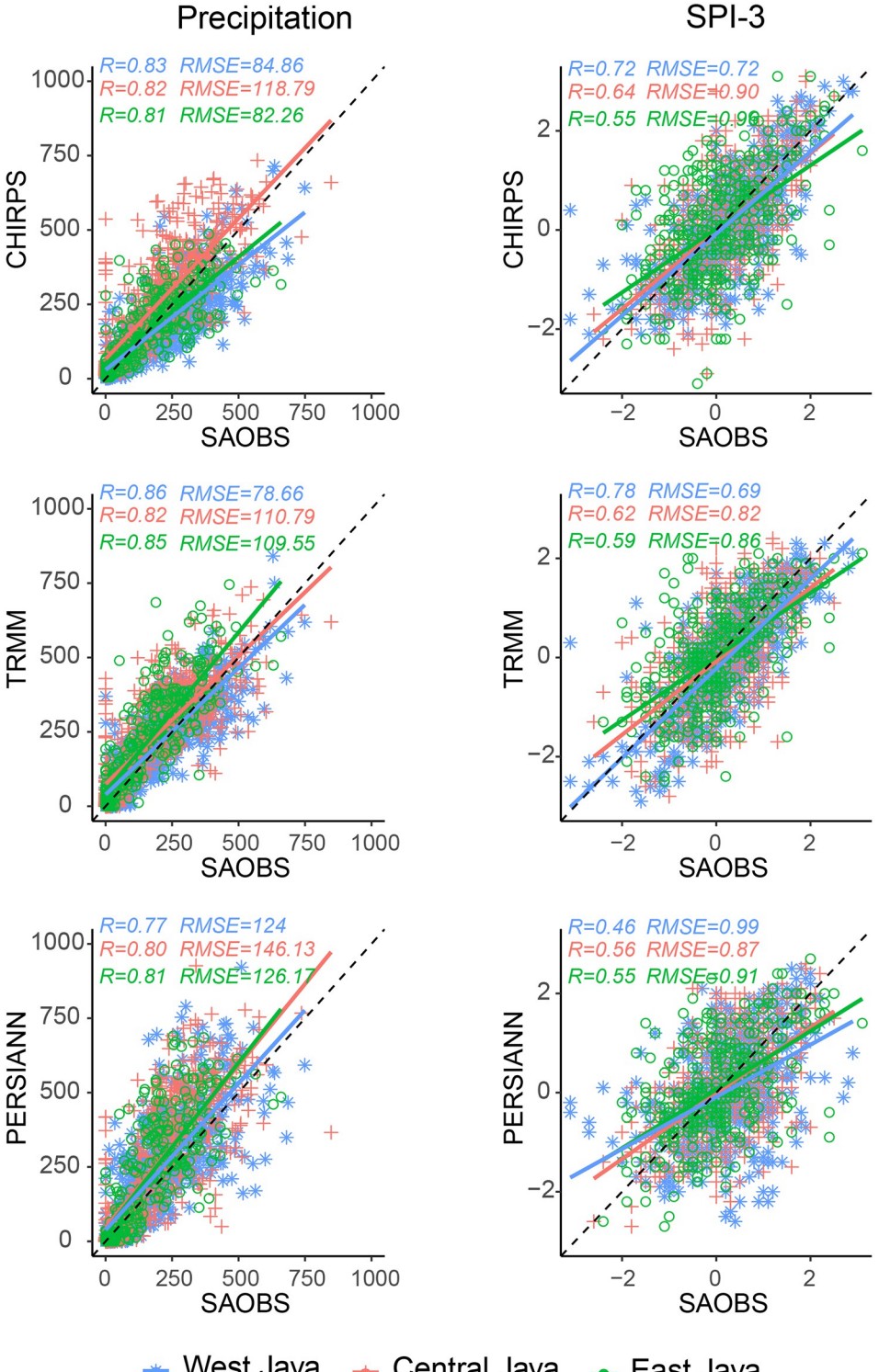

**Fig 6. Scatterplot of the regional pooled precipitation (left) and SPI-3 (right) for 2001–2017.** CHIRPS, TRMM, and PERSIANN were each plotted against SA-OBS.

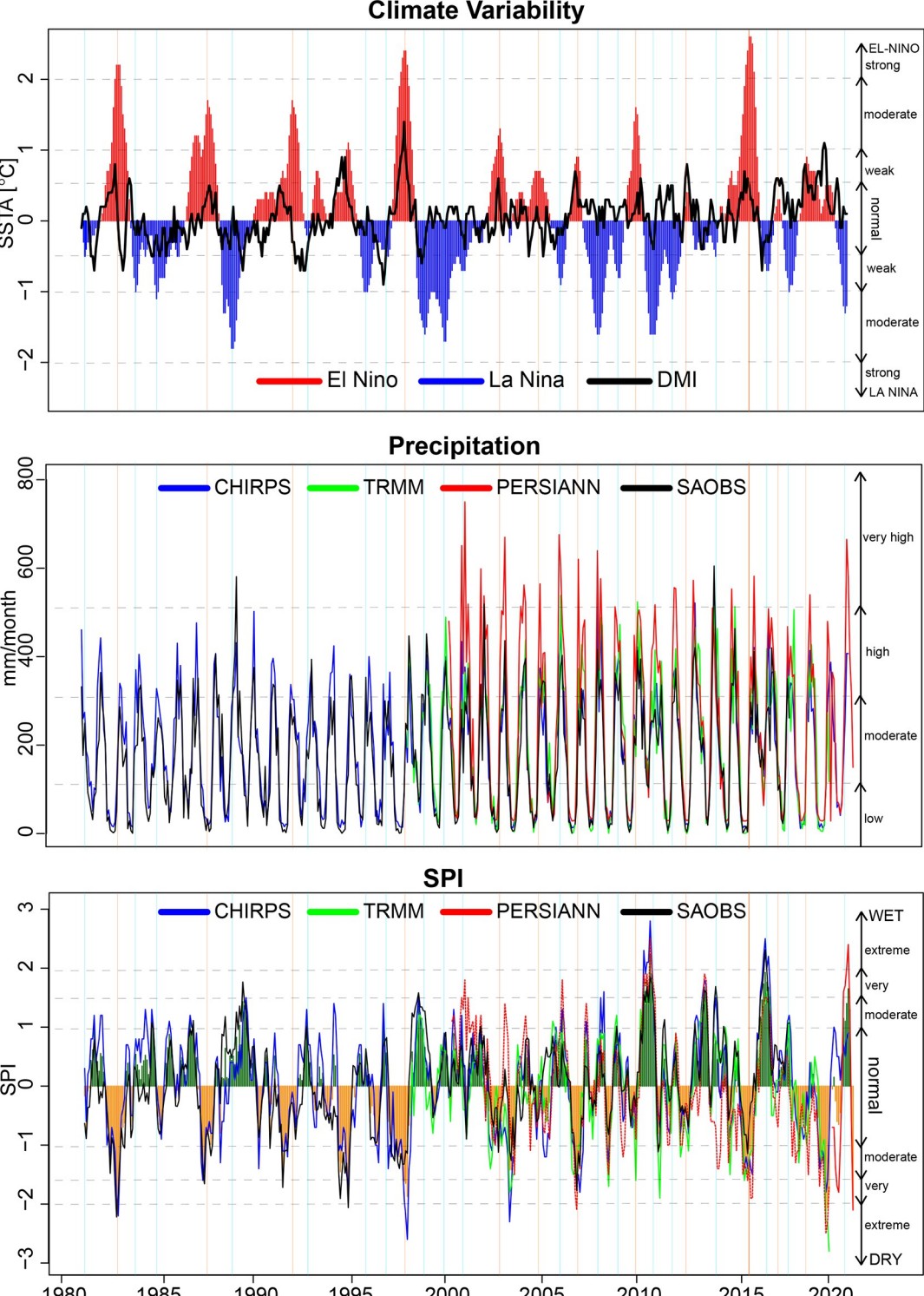

**Fig 7. Time series of the Oceanic Nino Index (ONI) and Dipole Mode Index (DMI) for climate events during 1981–2020 (top), the monthly precipitation derived from all datasets (middle), and SPIs (bottom).** For the SPI results, the colored bar shows the average from multiple sources, and the lines show the values derived from individual datasets: CHIRPS (blue), TRMM (green), PERSIANN (red), and SA-OBS (black). For each time series, the vertical orange solid line indicates a year with El Niño, and a vertical light-blue line indicates a year with La Niña. The horizontal gray dashed lines indicate the levels of ENSO (top), precipitation (middle), and SPI (bottom).

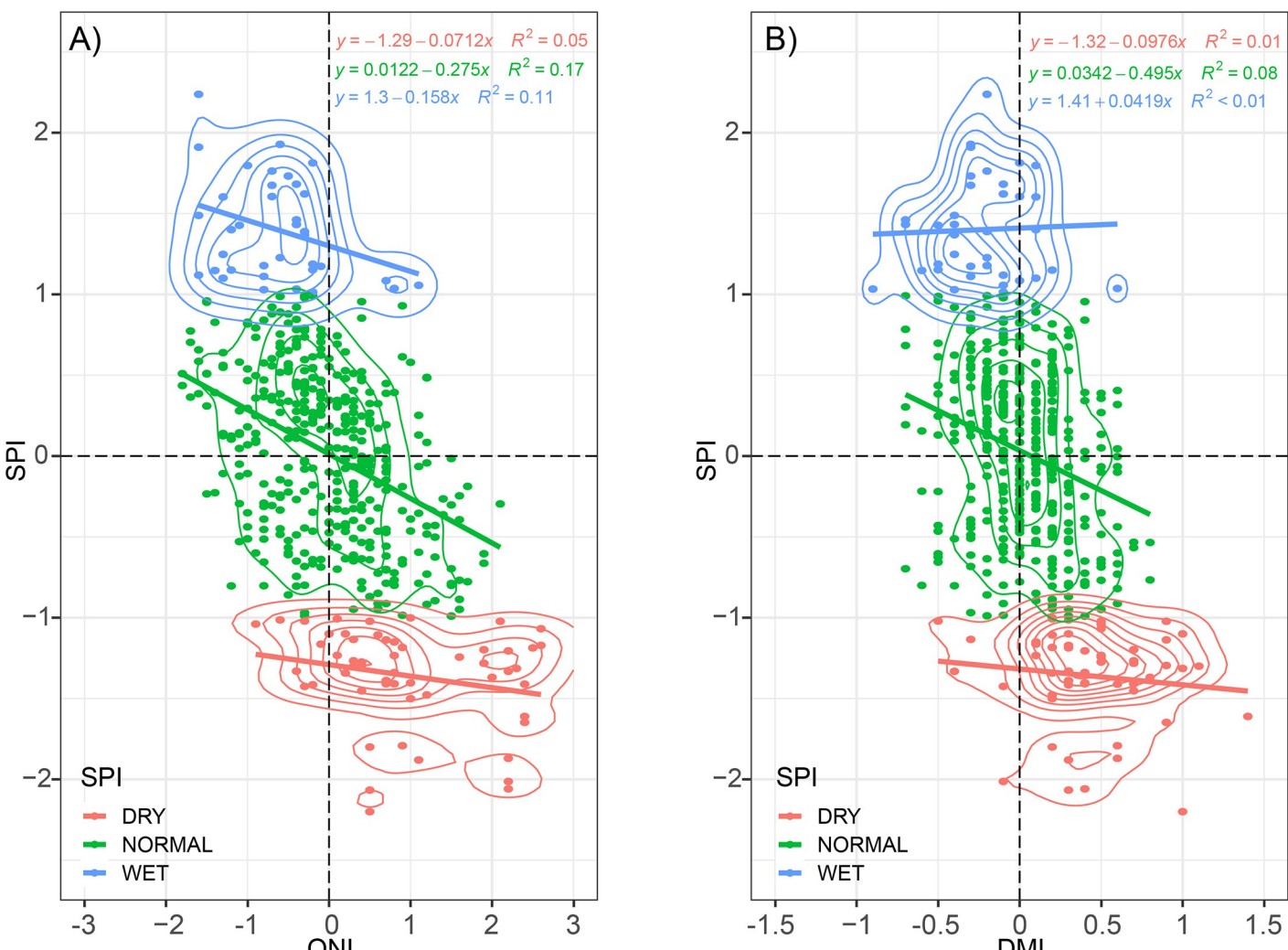

**Fig 8. Scatter and density plot of the (A) Oceanic Nino Index (ONI) and (B) Dipole Mode Index (DMI) against SPI classified as dry, normal, and wet conditions.** The SPI and its classification are determined from the averaged multiple source SPI as in Fig 7.

the El Niño events in 1982 and 1997. On the other hand, an extreme meteorological drought occurred in 2019, even though the El Niño had returned to neutral and was previously weak. The severe 2019 drought showed a stronger correlation with the positive-phase IOD, which was the strongest event according to historical records. It seems that the drought severity depends on the intensity of El Niño and positive-phase IOD occurring simultaneously, while the time phase seems to be modulated by the positive-phase IOD.

To summarize these results, the scatter and density plot between the averaged multisource SPI against the ENSO and IOD indices is shown in Fig 8. The linear correlation is then assessed within the three-class conditions according to SPI-based indicators: dry, normal, and wet. As expected, the dry condition has positive relations with the positive index of the ONI and DMI, with a slope of 0.07 and 0.09, respectively. For wet conditions, the effect of negative ONI (La Niña) is more significant than the effect of DMI. Moderate to very dry drought conditions consistently occur in the ONI and DMI threshold >0.5. Also, the intensity of wet and

dry SPI is not necessarily linear related to the magnitude of ONI and DMI. Both ONI and DMI have a significant role in the variability of the SPI during normal conditions.

## Discussion

In this study, we showed that the SPI-based meteorological drought indicator derived from satellite products may be applied to drought monitoring with more detail, wider coverage, and a faster timescale than ground observations. The multi-sources SPI-based meteorological drought indicator described well the relation of meteorological drought characteristics in Java island with El Niño and positive-phase IOD. This adds to a better understanding of the role of the Indian Ocean on drought variability in the study area. Several previous studies have characterized the Java Island drought which is influenced by El Niño in the Pacific Ocean [7, 8]. The use of satellite based-SPI as a meteorological drought has been used widely in many regions and many climatic zones. Suliman et al. in his study revealed a good consistency in SPI time series estimated using remotely sensed precipitation data and ground stations data, with the TRMM as the champion for monitoring droughts in different climatic zones of Iraq [59]. Not only a single source of satellite-based estimated rainfall, it is also possible to combine multi sources into merged satellite precipitation datasets [60]. A study by Rahman et al. (2021) found significantly improved performance of merged satellite d in monitoring the meteorological drought [59].

If the SPI or other meteorological drought indicators can be used to predict drought conditions in the next 3 months, then they can be used by the agricultural sector to anticipate and prepare for drought events in the near future. The satellite-based precipitation products can be utilized to predict future conditions including patterns, trends, and seasonal influences for water resource management at different timescales [60, 61]. This may be very useful for farmers, who need to make decisions on their planting strategies while accounting for the possibility of meteorological droughts that can progress to agricultural droughts. This is also important for water management authorities when estimating the water supply and planning a distribution strategy. The proposed satellite-based meteorological drought indicator can help farmers secure their economic livelihood and help the government guarantee food security. For example, rice production can be maintained by improved water management from several reservoirs and cloud seeding in anticipation of predicted meteorological drought events.

## Conclusion

SPI-based meteorological drought indicator was applied to multiple sources of satellite precipitation products for the largest rice-producing districts of Java, for which ground observations were still insufficient. The results showed that satellite products can be used to accurately describe the spatial and temporal distributions of meteorological drought events in the study area. A comparison with the ground observation dataset (SA-OBS) showed that the CHIRPS and TRMM had better correlations than PERSIANN of up to 0.6. Meteorological drought characteristics in the study area were shown to be strongly dependent on the variations in El Niño and the positive-phase IOD. The drought severity depends on the intensities of El Niño and positive-phase IOD occurring simultaneously, while the time phase seems to be modulated by the positive-phase IOD. Our results indicate the potential applicability of satellite-based precipitation monitoring to predicting meteorological drought events several months in advance, something valuable for an early warning. Further research is still needed to improve the satellite products accuracy by bias correction implementation, as well as investigating the time lag of meteorological drought which will have an impact on causing agricultural drought

in the study area. If this is provided, it will be very useful as a decision-making tool for farmers and the government to take action early.

The SPI can also be used to indicate wetness in the study area, which seems to be influenced by inter-annual climate variability in the Pacific Ocean and Indian Ocean (i.e., La Niña and negative-phase IOD), although these appear to have different temporal characteristics compared to meteorological droughts. Another study is needed to discuss this topic as it is outside the scope of the present study.

## Supporting information

**S1 File.**
(ZIP)

## Acknowledgments

We thank the (1) ESRI, (2) OpenStreetMap, (3) Agency for Meteorology, Climatology and Geophysics (BMKG), (4) NASA's Earth Data, (5) Center for Hydrometeorology and Remote Sensing (CHRS), (6) Climate Hazards Center UCSB, (7) Ministry of Agriculture Republic Indonesia, (8) National Weather Service Climate Prediction Center, (9) Global Climate Observing System, and (10) Department of Geography, Faculty of Mathematics and Natural Sciences, University of Indonesia for providing funds, data, and information of anything needed for this study. We would also thank the support team for the National Research Priority (PRN) Program: Drought Model Development for Sustainable Food Estate Management based on Nexus-Spatial Approach.

## Author Contributions

**Conceptualization:** Siswanto Siswanto, Andry Rustanto, Faris Zulkarnain, Muhammad Dimyati.

**Data curation:** Siswanto Siswanto, Kartika Kusuma Wardani, Andry Rustanto.

**Formal analysis:** Siswanto Siswanto.

**Funding acquisition:** Faris Zulkarnain, Muhammad Dimyati.

**Investigation:** Siswanto Siswanto, Kartika Kusuma Wardani, Babag Purbantoro, Andry Rustanto, Faris Zulkarnain, Muhammad Dimyati.

**Methodology:** Siswanto Siswanto, Kartika Kusuma Wardani.

**Project administration:** Kartika Kusuma Wardani, Faris Zulkarnain.

**Resources:** Siswanto Siswanto, Kartika Kusuma Wardani.

**Software:** Siswanto Siswanto, Kartika Kusuma Wardani.

**Supervision:** Siswanto Siswanto, Muhammad Dimyati.

**Validation:** Siswanto Siswanto.

**Visualization:** Siswanto Siswanto, Kartika Kusuma Wardani.

**Writing – original draft:** Siswanto Siswanto, Kartika Kusuma Wardani, Babag Purbantoro, Andry Rustanto, Faris Zulkarnain, Evi Anggraheni, Ratih Dewanti, Triarko Nurlambang, Muhammad Dimyati.

**Writing – review & editing:** Siswanto Siswanto, Kartika Kusuma Wardani.

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

     Index to Assess Drought Characteristics of the Kaoping River Basin, Taiwan. Water Resour. 2019;
     46(5).

52.  Lloyd-Hughes B, Saunders MA. A drought climatology for Europe. Int J Climatol. 2002; 22(13).

53.  WMO. Standardized Precipitation Index User Guide. J Appl Bacteriol. 2012; 63(3):197–200.

54.  Stagge JH, Tallaksen LM, Xu CY, Van Lanen HAJ. Standardized precipitation-evapotranspiration index
     (SPEI): Sensitivity to potential evapotranspiration model and parameters. In: IAHS-AISH Proceedings
     and Reports. 2014.

55.  James Adams. Climate Indices, an open source Python library providing reference implementations of
     commonly used climate indices [Internet]. 2017. https://climate-indices.readthedocs.io/en/latest/

56.  NCAR Command Language (n.d.) NCAR Command Language website, version 6.6.2. https://www.ncl.
     ucar.edu/. Accessed 8 January 2021.

57.  The R Project for Statistical Computing (n.d.) R Project website, version 3.6.3. https://www.r-project.
     org/. Accessed 12 February 2021.

58.  ArcGIS Desktop (n.d.) ESRI website, version 10.6. https://www.esri.com/en-us/home. Accessed 26
     December 2020.

59.  Rahman KU, Shang S, Zohaib M. Assessment of merged satellite precipitation datasets in monitoring
     meteorological drought over pakistan. Remote Sens. 2021; 13(9):1–37.

60.  Son B, Im J, Park S, Lee J. Satellite-based Drought Forecasting: Research Trends, Challenges, and
     Future Directions. Korean J Remote Sens. 2021; 37(4):815–31.

61.  AghaKouchak A, Farahmand A, Melton FS, Teixeira J, Anderson MC, Wardlow BD, et al. Remote sens-
     ing of drought: Progress, challenges and opportunities.  Vol. 53, Reviews of Geophysics. 2015.

