## [Decision Letter · Decision Letter 0]

3 Sep 2021

PONE-D-21-26705

Meteorological drought indicator based on multisource satellite data and the Standardized Precipitation Index: Application to Java Island in support of food security

PLOS ONE

Dear Dr. Dimyati,

Thank you for submitting your manuscript to PLOS ONE. After careful consideration, we feel that it has merit but does not fully meet PLOS ONE’s publication criteria as it currently stands. Therefore, we invite you to submit a revised version of the manuscript that addresses the points raised during the review process.

We look forward to receiving your revised manuscript.

Kind regards,

Shamsuddin Shahid

Academic Editor

PLOS ONE

Journal Requirements:

When submitting your revision, we need you to address these additional requirements. 1. Please ensure that your manuscript meets PLOS ONE's style requirements, including those for file naming. The PLOS ONE style templates can be found at https://journals.plos.org/plosone/s/file?id=wjVg/PLOSOne_formatting_sample_main_body.pdf and https://journals.plos.org/plosone/s/file?id=ba62/PLOSOne_formatting_sample_title_authors_affiliations.pdf
 2. Please update your Methods section to provide URLs where each of the 4 datasets used in the study can be accessed. 3. Please amend either the title on the online submission form (via Edit Submission) or the title in the manuscript so that they are identical. 4. We note that Figures 1 to 4 in your submission contain [map/satellite] images which may be copyrighted. All PLOS content is published under the Creative Commons Attribution License (CC BY 4.0), which means that the manuscript, images, and Supporting Information files will be freely available online, and any third party is permitted to access, download, copy, distribute, and use these materials in any way, even commercially, with proper attribution. For these reasons, we cannot publish previously copyrighted maps or satellite images created using proprietary data, such as Google software (Google Maps, Street View, and Earth). For more information, see our copyright guidelines: http://journals.plos.org/plosone/s/licenses-and-copyright. We require you to either (1) present written permission from the copyright holder to publish these figures specifically under the CC BY 4.0 license, or (2) remove the figures from your submission: a. You may seek permission from the original copyright holder of Figures 1 to 4 to publish the content specifically under the CC BY 4.0 license.   We recommend that you contact the original copyright holder with the Content Permission Form (http://journals.plos.org/plosone/s/file?id=7c09/content-permission-form.pdf) and the following text:“I request permission for the open-access journal PLOS ONE to publish XXX under the Creative Commons Attribution License (CCAL) CC BY 4.0 (http://creativecommons.org/licenses/by/4.0/). Please be aware that this license allows unrestricted use and distribution, even commercially, by third parties. Please reply and provide explicit written permission to publish XXX under a CC BY license and complete the attached form.” Please upload the completed Content Permission Form or other proof of granted permissions as an "Other" file with your submission. In the figure caption of the copyrighted figure, please include the following text: “Reprinted from [ref] under a CC BY license, with permission from [name of publisher], original copyright [original copyright year].” b. If you are unable to obtain permission from the original copyright holder to publish these figures under the CC BY 4.0 license or if the copyright holder’s requirements are incompatible with the CC BY 4.0 license, please either i) remove the figure or ii) supply a replacement figure that complies with the CC BY 4.0 license. Please check copyright information on all replacement figures and update the figure caption with source information. If applicable, please specify in the figure caption text when a figure is similar but not identical to the original image and is therefore for illustrative purposes only.The following resources for replacing copyrighted map figures may be helpful: USGS National Map Viewer (public domain): http://viewer.nationalmap.gov/viewer/The Gateway to Astronaut Photography of Earth (public domain): http://eol.jsc.nasa.gov/sseop/clickmap/Maps at the CIA (public domain): https://www.cia.gov/library/publications/the-world-factbook/index.html and https://www.cia.gov/library/publications/cia-maps-publications/index.htmlNASA Earth Observatory (public domain): http://earthobservatory.nasa.gov/Landsat: http://landsat.visibleearth.nasa.gov/USGS EROS (Earth Resources Observatory and Science (EROS) Center) (public domain): http://eros.usgs.gov/#Natural Earth (public domain): http://www.naturalearthdata.com/

Reviewers' comments:

Reviewer's Responses to Questions

**Comments to the Author**

1. Is the manuscript technically sound, and do the data support the conclusions?

Reviewer #1: Yes

Reviewer #2: Yes

2. Has the statistical analysis been performed appropriately and rigorously? 

Reviewer #1: No

Reviewer #2: Yes

3. Have the authors made all data underlying the findings in their manuscript fully available?

Reviewer #1: Yes

Reviewer #2: Yes

4. Is the manuscript presented in an intelligible fashion and written in standard English?

Reviewer #1: Yes

Reviewer #2: Yes

5. Review Comments to the Author

Reviewer #1: Dear Authors,

Please check these points:

1- Your manuscript contains the phrase “food security” actually I could not find any information about the main agricultural crop which is the rice. You should give information about the yield of the 8 selected regions which used for rice planting, in normal years, wet years and dry years. And how much is the losses in the yield because of drought events?

2- You cannot use only R values for the comparison among the different precipitation data or among the SPI data, better to use other statistical indexes to get more accurate results like the RMSE, RMAD, ME, and BIAS in addition to the R values.

3- for the keywords better to use only 5 keys, I would prefer to use keys as:

Meteorological drought, Satellite- based precipitation, SPI, Java Island, El-Nino

4- You cannot say that you have developed a meteorological drought indicator !! the use of SPI with precipitation derived from satellite data sources has been used widely .. for example

Awchi, T., and Suliman, A. H. A. (2021) Spatiotemporal assessment of meteorological drought using satellite-based precipitation data over Iraq. IOP Conf. Series: Earth and Environmental Science 779, 012052 IOP Publishing, doi:10.1088/1755-1315/779/1/012052

Suliman, A. H. A., Awchi, T. A., Al-Mola, M., & Shahid, S. (2020). Evaluation of remotely sensed precipitation sources for drought assessment in Semi-Arid Iraq. Atmospheric Research, 105007. https://doi.org/10.1016/j.atmosres.2020.105007

5- In line 178 , it is 48 months not 49

6- in line 387, give a number for citation of (Murat et al 2018)

Thanking you

Reviewer #2: The Editor,

Plos One.

Please find attached my comments, on the manuscript entitled “Meteorological drought indicator based on multisource satellite data and the Standardized Precipitation Index: Application to Java Island in support of food security

” submitted for publication in Plos One. I have made some comments (see attached file) and believe that they will be addressed by the authors.

It is a great pleasure to assess this paper. I have identified some issues that require modification in the text for improvement before the paper can be reconsidered for publication in Plos One or elsewhere.

The paper finds a relevant and important or topical issue in the ecological and environmental issues especially in meteorological drought appraisal using satellite data and the Standardized Precipitation Index. The scope of the paper is good for publication. The gap to be filled is not well-stated, this and other issues raised should be addressed during revision.

Thank you

6. PLOS authors have the option to publish the peer review history of their article (what does this mean?). If published, this will include your full peer review and any attached files.

Reviewer #1: **Yes: **Taymoor A. Awchi

Reviewer #2: **Yes: **Dr Israel Ropo Orimoloye

---

## [Author Response · Author response to Decision Letter 0]

8 Nov 2021

We would like to thank the editor and reviewers for his/her valuable comments. We have revised the manuscript and we hope that all comments and suggestions have been addressed and considered accordingly.

---

## [Decision Letter · Decision Letter 1]

22 Nov 2021

Satellite-based meteorological drought indicator to support food security in Java Island

PONE-D-21-26705R1

Dear Dr. Dimyati,

We’re pleased to inform you that your manuscript has been judged scientifically suitable for publication and will be formally accepted for publication once it meets all outstanding technical requirements.

Kind regards,

Shamsuddin Shahid

Academic Editor

PLOS ONE

Additional Editor Comments (optional):

Reviewers' comments:

Reviewer's Responses to Questions

**Comments to the Author**

1. If the authors have adequately addressed your comments raised in a previous round of review and you feel that this manuscript is now acceptable for publication, you may indicate that here to bypass the “Comments to the Author” section, enter your conflict of interest statement in the “Confidential to Editor” section, and submit your "Accept" recommendation.

Reviewer #1: All comments have been addressed

Reviewer #2: All comments have been addressed

2. Is the manuscript technically sound, and do the data support the conclusions?

Reviewer #1: Yes

Reviewer #2: (No Response)

3. Has the statistical analysis been performed appropriately and rigorously? 

Reviewer #1: Yes

Reviewer #2: (No Response)

4. Have the authors made all data underlying the findings in their manuscript fully available?

Reviewer #1: Yes

Reviewer #2: (No Response)

5. Is the manuscript presented in an intelligible fashion and written in standard English?

Reviewer #1: Yes

Reviewer #2: (No Response)

6. Review Comments to the Author

Reviewer #1: (No Response)

Reviewer #2: Dear Editor/Authors,

Authors have addressed comments from my side. I recommend the paper for publication in Plos One.

Thank you

7. PLOS authors have the option to publish the peer review history of their article (what does this mean?). If published, this will include your full peer review and any attached files.

Reviewer #1: **Yes: **Prof. Dr. Taymoor A. Awchi

Reviewer #2: **Yes: **Israel Ropo Orimoloye

---

## [Editor Report · Acceptance letter]

26 May 2022

PONE-D-21-26705R1 

Satellite-based meteorological drought indicator to support food security in Java Island 

Dear Dr. Dimyati:

I'm pleased to inform you that your manuscript has been deemed suitable for publication in PLOS ONE. Congratulations! Your manuscript is now with our production department. 

Kind regards, 

on behalf of

Dr. Shamsuddin Shahid 

Academic Editor

PLOS ONE